# Improving the Sensitivity of a Dark-Resonance Atomic Magnetometer

**DOI:** 10.3390/s25041229

**Published:** 2025-02-18

**Authors:** Hao Zhai, Wei Li, Guangxiang Jin

**Affiliations:** 1School of Instrumentation and Optoelectronic Engineering, Beihang University, Beijing 100191, China; 2Science and Technology on Vacuum Technology and Physics Laboratory, Lanzhou Institute of Physics, Lanzhou 730000, China; 3Hefei National Laboratory, Hefei 230088, China; 4Institute of Large-Scale Scientific Facility and Centre for Zero Magnetic Field Science, Beihang University, Beijing 100191, China; 5National Institute of Extremely-Weak Magnetic Field Infrastructure, Hangzhou 310051, China; 6Tianjin Quanmetry Technology Co., Ltd., Tianjin 300110, China; jgx@quanmetry.com

**Keywords:** magnetic field measurements within the geomagnetic range, experimental study, parameter optimization, quantum systems, dark-resonance atomic magnetometers

## Abstract

The combination of unmanned aerial vehicles and atomic magnetometers can be used for detection applications such as mineral resource exploration, environmental protection, and earthquake monitoring, as well as the detection of sunken ships and unexploded ordnance. A dark-resonance atomic magnetometer offers the significant advantages of a fully optical probe and omnidirectional measurement with no dead zones, making it an ideal choice for airborne applications on unmanned aerial vehicles. Enhancing the sensitivity of such atomic magnetometers is an essential task. In this study, we sought to enhance the sensitivity of a dark-state resonance atomic magnetometer. Initially, through theoretical analysis, we compared the excitation effects of coherent population trapping (CPT) resonance on the D_1_ and D_2_ transitions of ^133^Cs thermal vapor. The results indicate that excitation via the D_1_ line yields an increase in resonance contrast and a reduction in linewidth when compared with excitation through the D_2_ line, aligning with theoretical predictions. Subsequently, considering the impact of various quantum system parameters on sensitivity, as well as their interdependent characteristics, two experimental setups were developed for empirical investigation. One setup focused on parameter optimization experiments, where we compared the linewidth and contrast of CPT resonances excited by both D_1_ and D_2_ transitions; this led to an optimization of atomic cell size, buffer gas pressure, and operating temperature, resulting in an ideal parameter range. The second setup was employed to validate these optimized parameters using a coupled dark-state atom magnetometer experiment, achieving approximately a 10-fold improvement in sensitivity.

## 1. Introduction

Magnetic field detection technology operating within the range of the Earth’s magnetic field (excluding extremely weak magnetic fields such as those of the heart and brain that require shielding chambers) is of significant importance for applications such as mineral resource exploration, environmental protection, and earthquake monitoring, as well as the detection of sunken ships and unexploded ordnance (UXO) [1,2,3]. The combination of unmanned aerial vehicles (UAVs) and magnetic field measurement instruments is highly effective in magnetic field detection. SQUID-based magnetometers and magnetic field gradiometers are large in volume and weight and require liquid helium or liquid nitrogen for cooling, presenting difficulties in integration with UAVs. In such applications, numerous cases have employed cesium optical pumping magnetometers (OPMs) as the total field detection instrument for magnetic fields [1]. The sensitivity of an OPM is relatively high; however, due to the inherent characteristics in principle, there exists a measurement dead zone. The sensor probe has a radio frequency coil and is prone to electromagnetic interference, which is not conducive to the application of unmanned aerial vehicle (UAV) airborne magnetic field measurement.

Coherent population trapping (CPT) is a quantum interference phenomenon [4] that has been applied in atomic clocks [5] and magnetometers [6]. Among the many magnetic field measurement techniques [7,8], the Zeeman effect in respect of CPT in a small magnetic field (less than 1 mT) can be used to design atomic magnetometers, including atomic clocks based on single CPT or coupled dark-state atomic magnetometers using multiple CPT in parallel [9,10,11,12], which can be collectively referred to as dark-resonance atomic magnetometers.

The dark-resonance atomic magnetometer based on the CPT Zeeman effect is omnidirectional and possesses the ability to precisely measure the total field within the geomagnetic range [10,11,12]. It can be miniaturized and is suitable for applications in UAV-borne and portable magnetic field measurement. The dark-resonance magnetometer has the advantages of high precision and good long-term stability, unlike the optical pump magnetometer, which has no RF coil and therefore eliminates the systematic error caused by this coil [13]. The design of a sensor probe with full optical sensing can be realized. However, the dark-resonance atomic magnetometer has the problem of lower sensitivity compared to OPMs; therefore, it is very important to further improve the sensitivity of such atomic magnetometers. The sensitivity of a dark-resonance magnetometer is affected by many factors. The aforementioned factors are predominantly influenced by the parameters governing the quantum system. The wavelength of the radiation-source VCSEL laser is an important factor, and some parameters interact and restrict each other. Conducting theoretical analysis and parameter optimization experimental research on these influencing factors is an important technical approach to further improve the sensitivity of the dark-resonance atomic magnetometer.

In this study, we set up an experimental system and conducted research on optimizing the quantum system parameters of a coupled dark-state atomic magnetometer. Based on the relationship between the sensitivity of the atomic magnetometer and the CPT contrast and linewidth of the ^133^Cs D_1_ and D_2_ spectral lines, the performance of the CPT resonance line of the VCSEL was compared in respect of the ^133^Cs D_1_ and D_2_ spectral lines [14,15], and the influence of other parameters of the quantum system on the CPT resonance line performance was studied. The research methods and results provided in this paper can improve the sensitivity of dark-resonance atomic magnetometers.

This paper is organized as follows. Section 2 introduces the principle of the CPT atomic magnetometer based on the three-level Λ system and the Zeeman effect. It analyzes the factors influencing the CPT resonance linewidth, contrast, and sensitivity of the CPT atomic magnetometer, and comparatively examines the distinct impacts of D1 and D2 line excitations of ^133^Cs on these parameters. SymPy is utilized to compute the Clebsch–Gordan coefficients for different quantum parameters associated with the D1 and D2 lines, thereby validating the advantages of D1 line excitation. Section 3 outlines the physical design of the dark-resonance atomic magnetometer. Section 4 describes the experimental setups for parameter optimization and for coupling in the dark-state atomic magnetometer. Section 5 presents the experimental results along with their analysis. Section 6 summarizes the research findings.

## 2. Theory

### 2.1. The Three-Level Λ System

A coherent configuration with a narrow linewidth can trap resonances in a system with two long-lived states |1⟩ and |2⟩ and an excited state |3⟩ that is coupled by two light fields $E_{1}exp\left(-i\omega_{1}t-i\varphi_{1}\right)$ and $E_{2}exp\left(-i\omega_{2}t-i\varphi_{2}\right)$ to the excited state |3⟩, where E_1_ and E_2_ are the electric field amplitudes of the two laser beams respectively, while $\omega_{1}$ and $\omega_{2}$ are their corresponding angular frequencies, additionally, $\varphi_{1}$ and $\varphi_{2}$ are the additional phase of the optical fields, whose energy-level arrangement is similar to the Greek letter Λ for a three-level system and is therefore called the Λ system. For example, in alkali metal atoms, |1⟩ and |2⟩ can be associated with the two hyperfine components of the ground state, and |3⟩ and the first excited state are linked. Here, it is assumed that each light field interacts only with one transition. According to the definition, the double-photon detuning $\delta_{L}=\left(\Omega_{3}-\Omega_{2}\right)-\omega_{2}$ and the Raman detuning $\delta_{R}=\delta_{R}=\Delta_{21}-\left(\omega_{1}-\omega_{2}\right)$, $\Delta_{21}=\Omega_{2}-\Omega_{1}$ = $\Delta_{hfs}$ represent the splitting frequencies of the quantum states |1> and |2>. The energy levels are represented by the corresponding equivalent frequencies $\Omega_{i}={energy}_{i}/\hbar$ (where i = 1, 2, 3) as shown in Figure 1 [12,16].

The Raman frequency detuning $\delta_{R}$ can be considered as the frequency difference between the ground-state hyperfine structure splitting $\Delta_{21}=\Delta_{\mathrm{hfs}}$ (with a frequency $\Delta_{\mathrm{hfs}}$ of about 9.2 GHz in the alkali metal ^133^CS) and the frequency of a microwave generator $\nu_{RF}$. Dark resonance only appears in the vicinity of the very small frequency interval $\delta_{R}\;$ = 0 Hz. The energy changes in atomic levels under the influence of an external magnetic field (or the equivalent adjustments of frequencies $\omega_{1}$ and $\omega_{2}$ shown in Figure 1) are precisely reflected by the changes in $\delta_{R}$. Therefore, the sensitivity of measuring the magnetic field is directly related to the frequency interval of dark resonance appearing around $\delta_{R}\;$ = 0, and this characteristic is an important aspect of dark resonance for non-zero Raman frequency detuning $\delta_{R}\;$ ≠ 0.

The CPT coherent dark state can be prepared in the Λ system, where the two hyperfine levels of alkali metal atoms in the ground state are coupled to a common excited state by two laser fields that resonate with two atomic transitions. If the frequency difference between the laser fields is close to the atomic hyperfine splitting $\Delta_{\mathrm{hfs}}$ of the ground state, effective quantum coherence is produced between the two hyperfine components [17]. This effect can be detected as a dark line in fluorescence spectra, or as a resonant bright line in transmitted radiation. The CPT magnetometer uses the Zeeman effect of the magnetic field-sensitive ground-state transition (mF ≠ 0) to measure the magnetic field. Compared to the optical pumping magnetometer (OPM) that uses light intensity pumping, the physical scheme is more compact and can enable a fully optical sensor design and omnidirectional magnetic field measurement [18].

### 2.2. CPT Resonance Linewidth and Contrast

By solving the density matrix equation for the three-level Λ system atomic ensemble, one can obtain the linewidth formula for the CPT resonance [9,19,20]. The linewidth γ of the resonance line is the full width at half-maximum (FWHM) of the resonance spectral line.(1)$$\gamma =FWHM=\frac{1}{\pi }\left(\Gamma +\frac{\omega_{r}^{2}}{\Gamma^{*}}\right)$$

In this equation, Γ represents the coherent relaxation rate between the ground-state hyperfine levels of alkali atoms, which is given by $\Gamma =\Gamma_{w}+\Gamma_{bg}+\Gamma_{se}$, where $\Gamma_{w}$ is the collision relaxation between alkali atoms and the wall, $\Gamma_{bg}$ is the collision relaxation between alkali atoms and the buffer gas, and $\Gamma_{se}$ is the spin-exchange relaxation between alkali atoms. $\omega_{r}$ is the Rabi frequency at which the laser interacts with the alkali atoms, and $\Gamma^{*}$ is the relaxation rate of the excited state of alkali atoms.

The CPT signal is defined by contrast, which is the ratio of the signal amplitude to the background radiation, as shown in Equation (2), where the variables are the same as in Equation (1), C = Signal/DC Level, as shown in Figure 2. Contrast is a key parameter that affects the stability of CPT resonance or the sensitivity of magnetometers [21]. In the red spectral line shown in Figure 2, the central peak represents the CPT resonance signal, while the remaining portions correspond to the Doppler optical absorption line.(2)$$C=\frac{\omega_{r}^{2}/\Gamma^{*}}{\Gamma +\omega_{r}^{2}/\Gamma^{*}}$$

### 2.3. Sensitivity of CPT Resonance Magnetometer

Sensitivity is a key performance indicator for atomic magnetometers, and the sensitivity of the CPT resonance magnetometer can be expressed as follows [13,22,23,24]:(3)$$\delta B=\frac{1}{\gamma^{\prime }\sqrt{nTVt}}=\frac{1}{\gamma^{\prime }}kN\frac{\gamma }{S}$$
where $\gamma^{\prime }$ is the cesium atomic gyromagnetic ratio (3.5 Hz/nT), n is the atomic number density, T is the spin relaxation time, V is the measurement volume, t is the measurement time, the resonance factor k is approximately equal to 1, N is the root-mean-square (rms) noise value, S is the amplitude of the signal, and γ is the resonance linewidth, i.e., FWHM.

From Equation (3), it can be seen that increasing the atomic number density n, narrowing the resonance linewidth, increasing the signal amplitude (contrast), and increasing the relaxation time can improve the sensitivity of the CPT resonance magnetometer. However, each of these factors is mutually constraining. For example, increasing the signal amplitude and narrowing the linewidth are basically contradictory. A compromise can be achieved through experimental optimization of the parameters of the quantum system.

### 2.4. ^133^Cs Vapor CPT Resonance: D_1_ Versus D_2_ Line Excitation

Experiments using the D_2_ line of ^133^Cs have shown that the CPT signal contrast is approximately 1% [25,26], which limits the sensitivity of magnetometers [22], making it an important issue when designing quantum systems for magnetometers using the D_1_ and D_2_ lines of ^133^Cs (excitation from 6^2^S_1/2_ to 6^2^P_l/2_ and 6^2^P_3/2_, respectively).

One important difference between the two excitation lines of ^133^Cs is the excited state hyperfine structure, as shown in Figure 3. The P_1/2_ excited state has only two hyperfine levels (labeled F’ = I ± 1/2, where I is the nuclear spin quantum number 7/2), both of which are coupled to the two hyperfine ground states, used for coherent preparation. By comparison, the P_3/2_ excited state has four excited hyperfine levels (F’ = I ± 1/2, I ± 3/2), and according to the selection rules, only two of them (F’ = I ± 1/2) are simultaneously coupled to the two ground-state hyperfine levels. The uncoupled level (I ± 3/2) does not contribute to coherent preparation and reduces the lifetime of the coherent dark-state because it allows atoms to escape from the trapping state by direct (single-photon) absorption. Compared with the excitation of the D_1_ transition, this increases the CPT resonance width and reduces its contrast.

Another difference between the two excitations is the Clebsch–Gordan coefficients (C-G coefficients) for optical transitions; for example, in the case of the excited state, the excited light to some extent couples the ground state with the two excited state hyperfine levels, so each fine structure line will excite a pair of dark states simultaneously. If the ratio of the C-G coefficients is equal, then the coherence will be maximized, and the total resonance contrast of the dark lines will be higher. If the ratio of the C-G coefficients is not equal, the contrast will be reduced. For the ^133^Cs atom, the ratio of the C-G coefficients is equal for the two dark states in the D_1_ line excitation, but it is not equal for the two dark states in the D_2_ line excitation [27]. Based on the Python framework, the calculation program package “physics.quantum.cg” in SymPy can be introduced for simple calculation. Firstly, we define the parameters of the D_1_ line and D_2_ line (including nuclear spin quantum number I, total angular momentum quantum number J, total angular momentum quantum number F, etc.), and then traverse all single magnetic quantum numbers m_I_ and m_J_ that satisfy m_I_ + m_J_ = m_F_ [28]. Then, we create a Clebsch–Gordan object and use the doit() method to calculate the coefficients. By comparing the calculation results for the C-G coefficients of the corresponding quantum numbers of the D_1_ line and D_2_ line, it can be found that the C-G coefficients of the magnetic quantum number transitions corresponding to the D_1_ line are equal, while those of the D_2_ line are not equal. Therefore, theoretically, a higher contrast can be obtained for the D_1_ line excitation.

## 3. Design of a Dark-Resonance Atomic Magnetometer

In 1992, Scully and Fleischhauer theoretically proved that CPT resonance could be used in a sensitive magnetometer device [10,11]. The first experimental realization of a single CPT dark-state magnetometer was reported in 1998 [6]. Single dark-state CPT magnetometers (DSMs) and coupled dark-state magnetometers (CDSMs) based on multi-CPT resonance have been studied [6,12,29,30], and the sensitivity, which depends on the quantum system parameters such as resonance line width, is a key characteristic of such magnetometers, in addition to accuracy and long-term stability.

In the presence of a magnetic field, several CPT resonances appear in the form of a spectrum [12,31]. Ideally, the Zeeman frequency $v_{B}$ corresponding to the magnetic field B can be determined via the linear Breit–Rabi formula [28,32], which is expressed as a Taylor series expansion and neglects the quadratic and higher-order terms in B. The measurement principle remains valid when the precise form of the Breit–Rabi formula is applied.(4)$$\nu_{B}=\frac{\mu_{B}}{2\cdot \left(2I_{k}+1\right)h}\left[|n|\left(g_{J}-g_{I}\right)+8\Delta {mg}_{I}\right]B$$
where $\nu_{B}$ refers to the Zeeman frequency of the magnetic field B, $\mu_{B}$ to the Bohr magneton of ^133^Cs, $g_{J}$ to the Landé factor for the fine structure, $g_{I}$ to the Landé factor for the atomic nucleus, the label for the CPT dark resonance $n=m_{F1}+m_{F2}$ to the sum of the total magnetic quantum numbers $m_{F1}$ and $m_{F2}$, $I_{k}$ to the nuclean spin quantum number of the ^133^Cs, △m to the difference of $m_{F1}$ and $m_{F2}$, and h to the Planck constant.

Figure 4 shows the D₁ excitation scheme within the hyperfine structure of ^133^Cs. F denotes the total angular momentum quantum numbers, while the magnetic quantum numbers of the 6²S₁_/2_ ground state are denoted by m_F_; for the excited states 6²P_1/2_, the symbols are primed. The vacuum wavelength $\lambda_{FS}$ corresponds to the fine structure transition 6²S_1/2_→6²P_1/2_. The hyperfine ground-state splitting frequency is denoted by $\upsilon_{HFS}$. Each pair of blue arrows builds up a so-called Λ system, which causes a magnetic field-dependent CPT resonance. The gray-colored interrupted lines represent the frequencies corresponding to mF that are not affected by the magnetic field. The frequency between the upper and lower energy levels of the ground state is equal to $\upsilon_{HFS}$. The energy shift introduced by the magnetic field is expressed as $\nu_{B}$. According to the selection rules of two-photon transitions [19,29], there may be an n = 0 Λ system (magnetically insensitive), two n = ±1 Λ systems, four n = ±2 (displaying two in Figure 4) Λ systems, and four n = ±3 Λ systems; n = ±4 and n = ±6 Λ systems can be ignored when measured.

## 4. Experimental Setup

Two experimental setups were constructed, one of which was used to optimize the parameters of the dark-state resonance magnetometer quantum system, and the other was used to test the performance indicators of the dark-state resonance magnetometer.

These two setups are interrelated and both utilize the inherent characteristics of atomic transitions, particularly CPT resonance. The first setup measures the 0–0 transition spectral lines, thereby eliminating the influence of external magnetic fields. This enables more precise measurement of parameters such as linewidth and contrast. The second setup is based on the principle of dark-resonance atomic magnetometer and is used for performance verification after optimizing the parameters of the quantum system. In this configuration, it measures the frequency difference between the CPT resonance with non-zero magnetic quantum numbers and the 0–0 transition. Utilizing these two setups to enhance the sensitivity of the dark-resonance atomic magnetometer is an innovative method in this research.

### 4.1. Parameter Optimization Experimental Setup

The experimental setup for parameter optimization is shown in Figure 5. A dual-color light field is generated by modulating the injection current of the VCSEL laser at half of the hyperfine splitting frequency ($v_{HFS}\;$ = 9.2 GHz) to form the Λ system. The two first-order optical sidebands are completely hyperfinely split apart, forming the Λ system. The system uses an independent reference atomic cell through laser frequency modulation and a locking loop to stabilize the VCSEL laser frequency. The adjustment of the light intensity is achieved by using different transmission rate neutral filters to produce σ+ circularly polarized light with a quarter-wave plate. The atomic cell constant-temperature bath can be set to different working temperatures and controlled to an accuracy of 0.01 °C. The sweep signal is combined with a direct digital synthesizer (DDS) to achieve 4.6 GHz microwave frequency synthesis and Raman frequency scanning, producing a CPT resonance signal that is displayed on an oscilloscope and captured and processed by a PC. The oven-controlled crystal oscillator (OCXO) is the reference clock for the 4.6 GHz microwave frequency.

### 4.2. Coupled Dark-State Atomic Magnetometer Experimental Setup

The coupled dark-state atomic magnetometer experimental setup, as shown in Figure 6, is a principal device for a dark-resonance atomic magnetometer that conforms to the excitation scheme of the ^133^Cs hyperfine structure D_1_ line spectrum, as shown in Figure 6. The laser frequency stabilization control loop is the same as that in Figure 5. The Zeeman frequency generator generates a frequency that is associated with the magnetic field B, modulates the 4.6 GHz microwave signal, and causes the VCSEL laser to produce a second-order frequency-modulated spectrum. The orthogonal output signal of the lock-in amplifier is demodulated to obtain the CPT resonance peak, whose zero point corresponds to the frequency value $\nu_{B}$ in (4), and can be used to calculate the value of the magnetic field B using (4). The collected magnetic field B data are processed to produce a Fourier transform amplitude spectrum (i.e., the root mean square of the noise power spectrum), which can provide the power spectral density (PSD) curve, while the PSD value at 1 Hz is used as a reference value for the sensitivity of the magnetometer [23,33].

The lock-in amplifier utilized in this setup is the MFLI model from Zurich Instruments. The remaining electronic modules consist entirely of in-house developed circuit boards. These custom-designed boards have undergone rigorous testing and validation, ensuring that their signal-to-noise ratio and measurement accuracy meet the stringent requirements of our experiments. The general equipment depicted in Figure 5 and Figure 6 is identical.

## 5. Experimental Results and Analysis

The sensitivity of the dark-state resonance atomic magnetometer is associated with factors such as the D1 or D2 line transitions, laser power, transition signal contrast, linewidth, etc. Among these parameters of quantum systems, the properties of the D1 and D2 line VCSEL lasers, the temperature of the cesium atomic cell, the size of the cesium atomic cell, and the buffer gas pressure in the cell were intensively studied. We used the least squares method to fit the experimental data.

### 5.1. Comparison of D_1_ and D_2_ Line VCSEL Lasers

In the experimental setup shown in Figure 5, one of the two VCSEL lasers with different wavelengths was used, with one being adjusted to the D_1_ transition of ^133^Cs at 894.6 nm, and the other being adjusted to the D_2_ transition of ^133^Cs at 852.3 nm. Both lasers have the same 2 mm beam diameter and are $\sigma^{+}$ circularly polarized, so they are projected along the same path in the gas chamber and detected using a photodiode.

Line D_2_ has twice the photon degeneracy of Line D_1_, with 894 nm and 852 nm VCSELs having a single longitudinal cavity mode and single transverse cavity mode, with stable polarization over a wide tuning range, a high modulation bandwidth, and requiring approximately 1 mW of an RF signal to generate a CPT resonance signal for probing to produce the optimal sideband. The VCSEL must have high conversion efficiency, a threshold current of less than 1 mA, and power consumption of less than 2 mW, and be able to generate more than 100 μW of optical power.

The atomic cell has a length of 20 mm and contains a mixture of ^133^Cs and N2/Ar buffer gas. A magnetic shield is placed outside the atomic cell to shield it from external magnetic fields. A small longitudinal magnetic field (about 10 μT) is applied using a magnetic field coil to keep the magnetically sensitive CPT resonance away from the magnetically insensitive CPT resonance (|F = 3, mF = 0>→|F = 4, mF = 0>). The diode laser frequency is locked to the optical transition center using a laser frequency lock loop. To measure the CPT resonance, a DDS and sweep signal source generate microwave frequencies and scan a few kHz around 4.6 GHz.

The absorption spectrum of VCSEL after passing through an atomic cell is shown in Figure 7. The blue line represents the D2 line laser, and the red line represents the D1 line laser. After the VCSEL is modulated by a 4.6 GHz microwave, the peak-to-peak increase in the absorption peak is approximately five times, and the trace is normalized to the Doppler absorption value.

Comparing the D_1_ and D_2_ transition excitation for the CPT resonance, the linewidth of the CPT resonance excited by the D_1_ line is smaller, and the contrast is larger. The contrast excited by the D_1_ line is close to 10%, while the contrast excited by the D_2_ line is close to 2%. For an atomic cell with the same process parameters (cell length, buffer gas pressure, etc.), we independently optimized the atomic cell temperature for each of the two transitions to obtain the best contrast CPT resonance peak. The contrast and linewidth of the signals for the two cases are compared in Figure 8, with the blue line representing the measured data and the red line representing the Lorentzian fitted spectral line.

It can be seen from (3) that the sensitivity can be estimated by the linewidth and the contrast ratio (the reciprocal of the slope of the dark-resonance absorption line center), as shown in Figure 8. Using the D_1_ transition, the slope of the dark-resonance center can be obtained almost one order of magnitude larger than that obtained using the D_2_ transition, which indicates a significant improvement in atomic clock or magnetometer performance.

Laser power broadening is an important mechanism that affects the linewidth of the CPT resonance peak. Under the same VCSEL laser power, the linewidth of the D_1_ line is smaller than that of the D_2_ line, and the D_1_ line and D_2_ line are related to the VCSEL power by different exponential relationships. After fitting, the relationship between the D_1_ line γ and VCSEL power is γ ∝ P^0.5^, and the relationship between the D_2_ line γ and VCSEL power is γ ∝ P^0.6^. The relationship between the CPT resonance linewidth and VCSEL wavelength and laser power is shown in Figure 9. Due to power broadening, the CPT linewidth increases with the laser power. The relationship between the contrast of the CPT resonance and the VCSEL wavelength and laser power is shown in Figure 10. The contrast also varies with the incident laser power, with a peak, and the contrast has already decreased significantly at laser powers above 100 μW. The incident laser power into the atomic cell can be set to around 20 μW.

### 5.2. Laser Noise

The noise power spectrum of the D1 laser is shown in Figure 11. The noise is linearly related to the DC current of the laser and has a significant impact on sensitivity. The loop noise composed of OCXO noise, DDS noise, preamplifier, and lock-in amplifier in the electronic system is not further experimentally analyzed in this paper.

### 5.3. Optimization of Other Parameters in Quantum Systems

The size of the atomic cell, the buffer gas pressure, and the working temperature are also important factors affecting the CPT resonance signal. We conducted a study on the influence of the atomic cell size on the CPT signal, measuring the linewidth and contrast of the CPT signal after the light beam passed through atomic cells of different sizes. The test showed that when the size of the atomic cell increased, the contrast of the CPT signal increased while the linewidth decreased. We prepared samples of cesium atomic cells of different sizes, as shown in Figure 12. The relationship between the contrast and linewidth of the CPT signal obtained experimentally with different sizes of cesium cells under the same light field is shown in Figure 13. The black curve in Figure 13 illustrates the relationship between contrast and gas cell size, while the red curve depicts the dependence of CPT resonance linewidth on cell size. Under the action of the same light field, the longer the cesium atomic cell length L, the higher the contrast of the CPT signal and the narrower the linewidth. However, the size of the atomic cell is constrained by design, and in order to meet the requirements of miniaturization design while taking into account the influence of atomic cell temperature on the CPT signal, we expect a contrast ratio of 7% and a linewidth of approximately 350 Hz. After optimizing these parameters, we selected the optimal size of the cesium atomic cell at 15 mm, while the diameters of all the cells are all 13 mm.

The operating temperature of the atomic cell affects the contrast of the CPT signal. When the temperature rises, the atomic density increases, and the CPT signal will accordingly be enhanced. However, the noise will also increase accordingly, and the signal-to-noise ratio is not proportional to the temperature in a simple way. At the same working temperature of the atomic cell, the laser power incident on the atomic cell is different, and the contrast of the CPT signal is also different. The experimental results are shown in Figure 14. From the measured data of the CPT signal contrast versus laser power at four different temperature points, as shown in the figure, it can be seen that when the temperature is higher, the contrast of the CPT signal changes slowly with the intensity of the light, and the contrast in the weak light power region increases rapidly with the decrease in temperature. When the temperature is lower than a certain value, the contrast of the CPT signal reaches a maximum value at the light power curve. Notably, if the temperature is too low, even if the contrast is large, when the CPT signal amplitude is too small, the signal-to-noise ratio will deteriorate. From the experimental results, the operating temperature of the cesium atomic cell is preferably in the range of 40 °C to 50 °C.

In an atomic cell without a buffer gas, the linewidth of the CPT resonance line is mainly determined by the transition time. When a certain pressure of buffer gas is introduced, the linewidth can be greatly reduced, thereby narrowing the linewidth. Inert gasses such as Ne, N_2_, and Ar are often used as buffer gasses to reduce the linewidth of the CPT signal [21]. The frequent collisions between alkali metal atoms and the buffer gas increase the diffusion time of alkali metal atoms to the wall of the cell, resulting in an increase in the coherence lifetime of the alkali metal atoms in the ground state and a decrease in the coherence relaxation rate, thereby reducing the CPT resonance linewidth. However, due to frequent collisions between buffer gas molecules and alkali metal atoms, although most of the collisions between ground-state alkali metal atoms and buffer gas molecules are elastic collisions, the collision process will perturb the wave function of the alkali metal atom’s motion state to some extent, causing a small shift in the energy level, which will produce a frequency shift, known as a collision frequency shift. The collision frequency shift is proportional to the buffer gas pressure in the cell at the same temperature. It is linearly related to the volume of the cell at the same temperature. The parameters describing the relationship between the collision frequency shift and the buffer gas pressure are the pressure frequency shift coefficient (α), with units of Hz/Torr, and the temperature frequency shift coefficient (δ), with units of Hz/Torr °C. Table 1 shows the pressure frequency shift coefficients (α) and temperature frequency shift coefficients (δ) of several commonly used buffer gasses [34]. Both α and δ have positive and negative values, and appropriate combinations can be used to ensure that the linewidth is narrowed while the temperature coefficient is close to zero.

When buffer gasses were introduced into the atomic cell, in order to reduce the collision broadening, we selected N_2_ with a positive pressure shift coefficient and Ar with a negative pressure shift coefficient, with a pressure ratio of 1:2. In the experiment, the total pressure of N_2_ and Ar was 10 Torr to 30 Torr, and the linewidth of the CPT signal obtained by interacting with different incident laser powers was measured. The results show that when the pressure of the buffer gas in the atomic cell is increased, there is a trend for the CPT signal linewidth to narrow, while when the incident laser power is increased, there is a trend for the CPT signal linewidth to widen, as shown in Figure 15, in which the curve characteristics are similar to those in Figure 9. Therefore, it is necessary to compromise and optimize the buffer gas pressure ratio, gas pressure, and light intensity parameters. As a result, the total pressure of N_2_ and Ar in the cell was 30 Torr while adjusting the laser intensity and temperature points to meet the requirements of optimization.

### 5.4. Results

In the coupled dark-state magnetometer experimental setup shown in Figure 6, the 894 nm VCSEL laser from the D_1_ line was used, and the magnetic field sensitivity measurement method described in the previous section was used. After optimizing the quantum system parameters, the magnetic field measurement sensitivity of the coupled dark-state magnetometer before and after the improvement was compared, as shown in Figure 16. The noise power spectrum at 1 Hz was improved from 20 pT/√Hz to less than 2 pT/√Hz, an improvement of more than ten times. By comparison, a detection noise of 50 pTrms at a 1 s integration time had previously been achieved in the coupled dark-state magnetometer of the China Seismo-Electromagnetic Satellite [29].

## 6. Discussion and Conclusions

The excitation of the D1 transition led to a reduction in the resonance linewidth and an increase in the resonance contrast, resulting in a significant improvement in the sensitivity of the coupled dark-state magnetometer using the D1 transition VCSEL laser. Based on this observation, parameter optimization experiments enabled us to identify suitable quantum parameters such as light intensity, buffer gas pressure, ratio in an atomic cell, and temperature of an atomic cell. We then further optimized the linewidth and contrast to improve the sensitivity of the coupled dark-state magnetometer by more than ten times.

The sensitivity of the coupled dark-state magnetometer is affected by a variety of factors, as can be seen from the experimental data. The stability of the laser power is crucial for the sensitivity of such magnetometers, and power fluctuations will directly result in a decrease in sensitivity. Therefore, adopting a high-power stable laser source is an effective way to improve the sensitivity. Electronic noise is another important aspect that affects sensitivity, and further research is needed in this area.

## Figures and Tables

**Figure 1 sensors-25-01229-f001:**
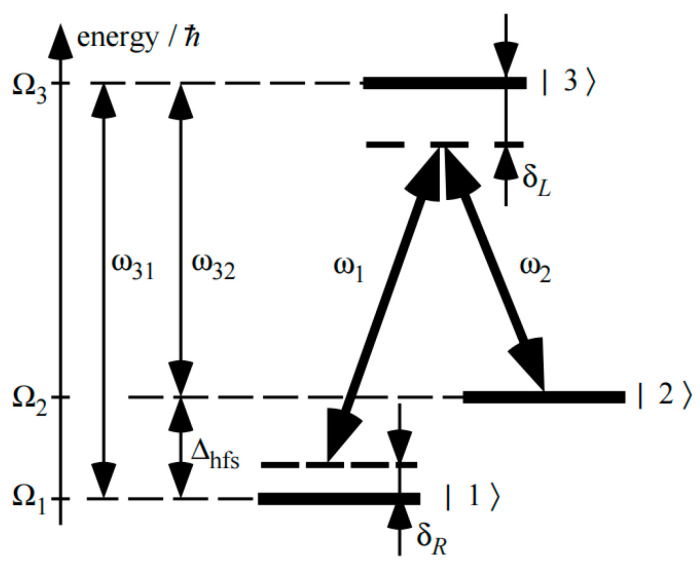
Three-level Λ system.

**Figure 2 sensors-25-01229-f002:**
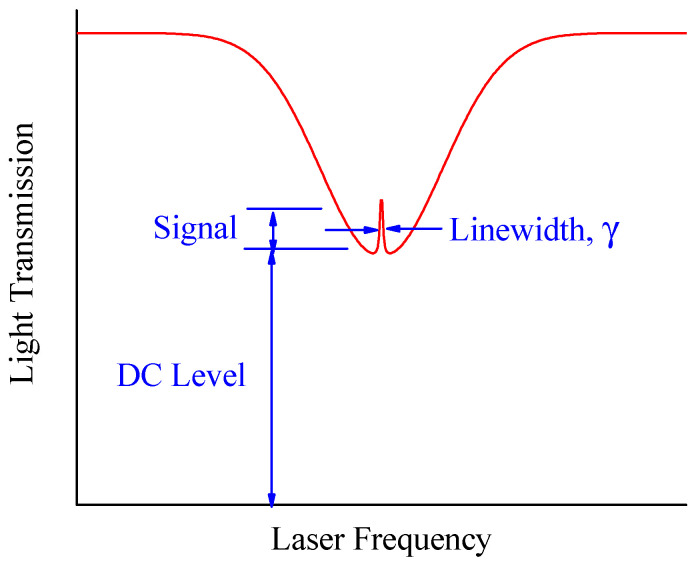
Contrast of CPT resonance.

**Figure 3 sensors-25-01229-f003:**
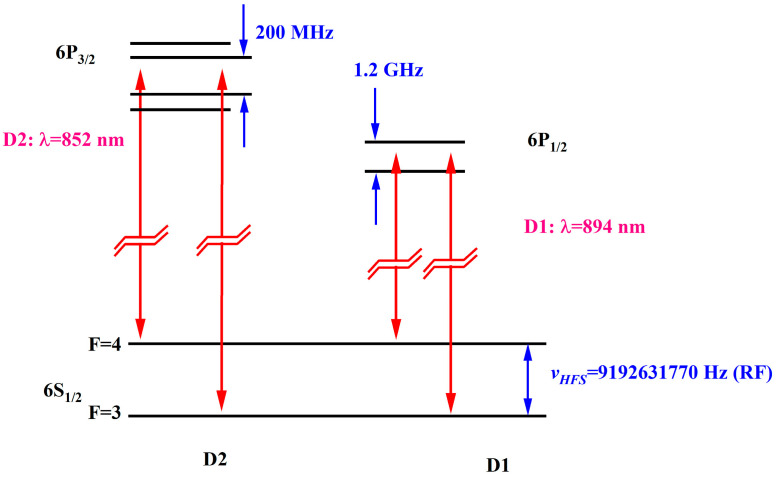
D_1_ and D_2_ lines of ^133^Cs.

**Figure 4 sensors-25-01229-f004:**
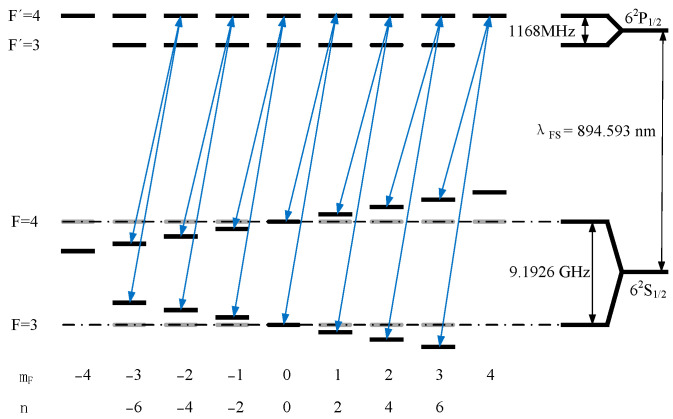
D₁ spectral line excitation scheme within the hyperfine structure of ^133^Cs.

**Figure 5 sensors-25-01229-f005:**
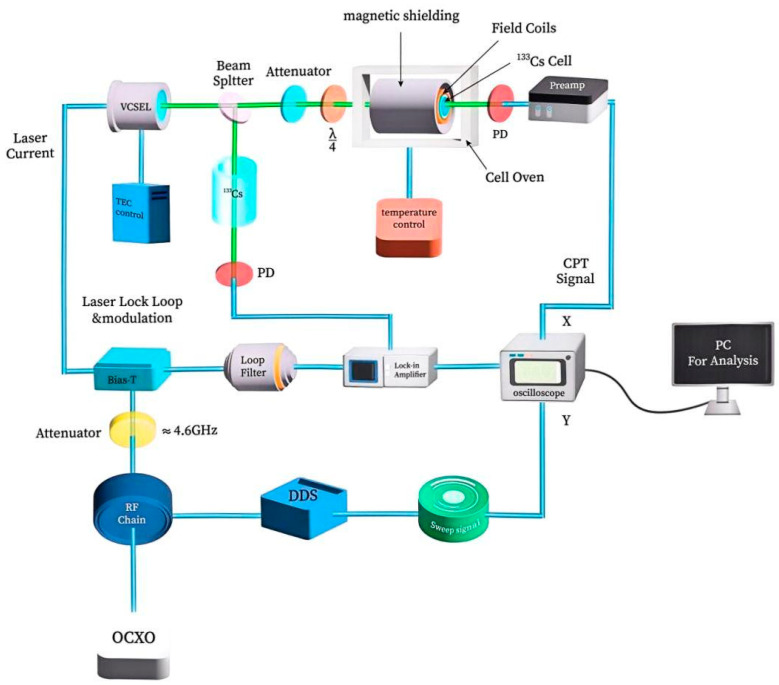
Parameter optimization experimental setup.

**Figure 6 sensors-25-01229-f006:**
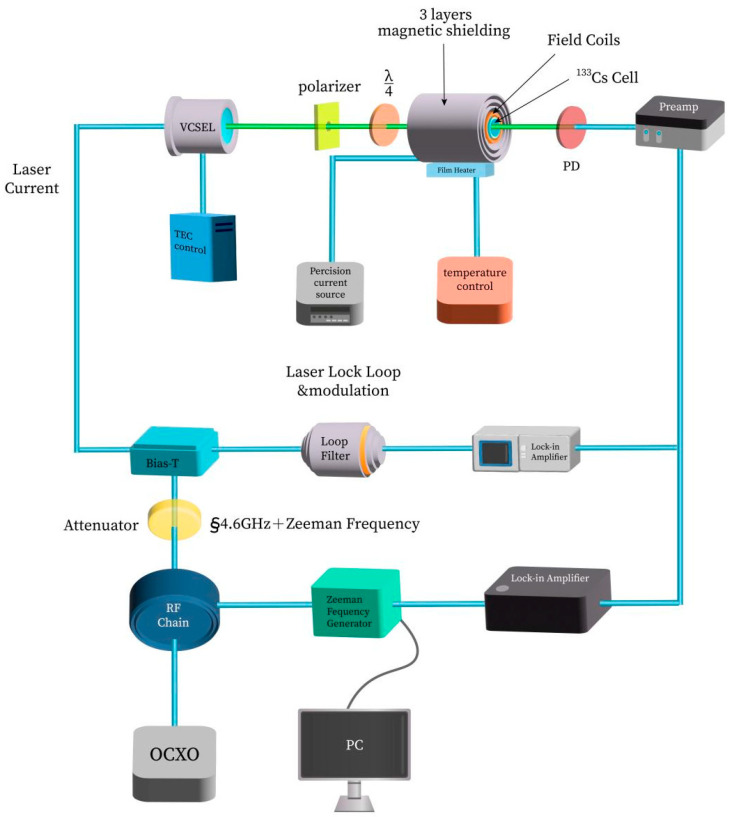
Coupled dark-state atomic magnetometer experimental setup.

**Figure 7 sensors-25-01229-f007:**
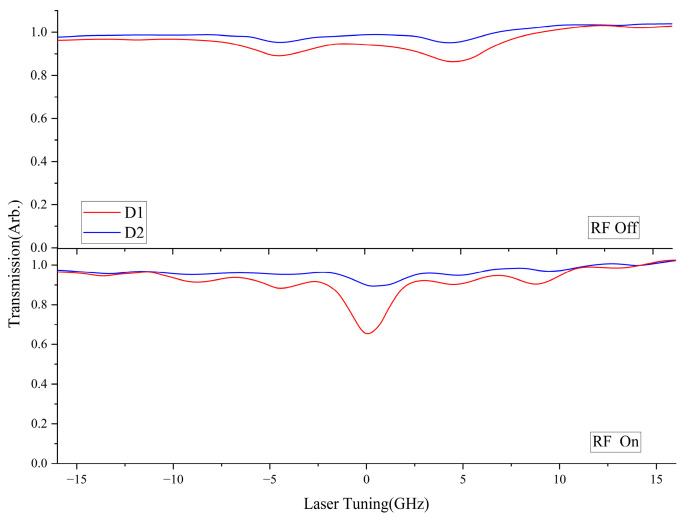
Doppler absorption spectrum of VCSEL laser after passing through atomic gas cell.

**Figure 8 sensors-25-01229-f008:**
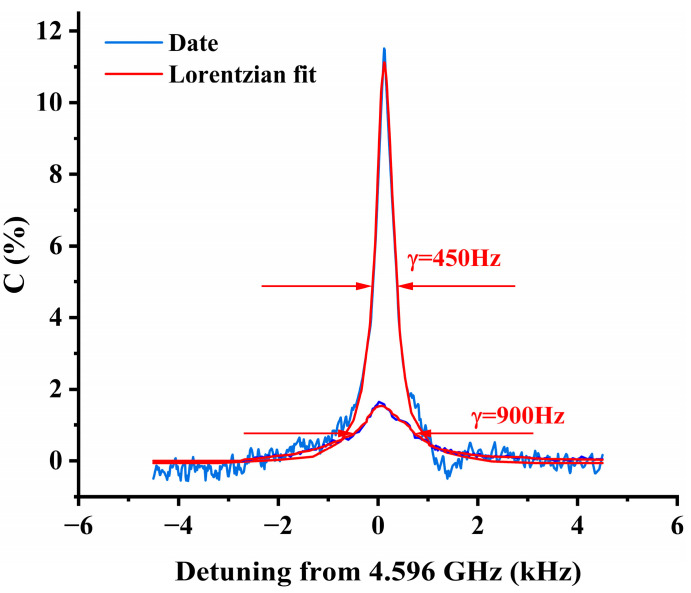
Linewidth and contrast in cell: D_1_ and D_2_.

**Figure 9 sensors-25-01229-f009:**
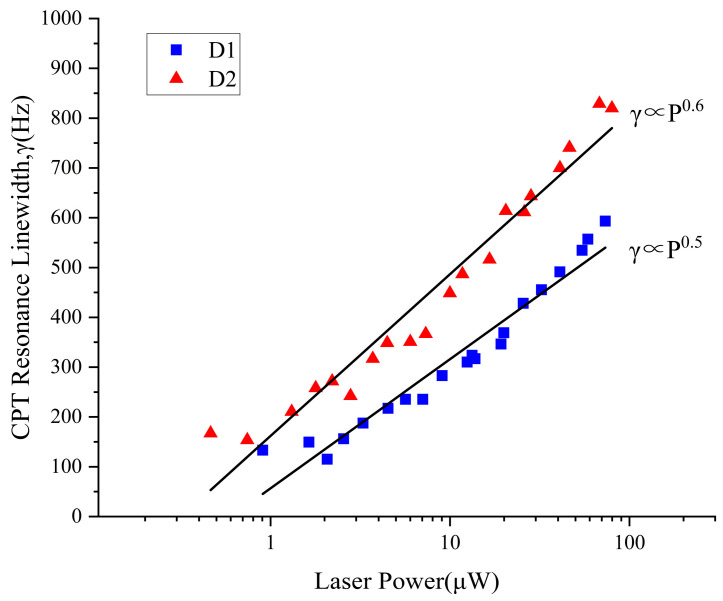
The CPT resonance linewidth is related to the wavelength and laser power of the VCSEL.

**Figure 10 sensors-25-01229-f010:**
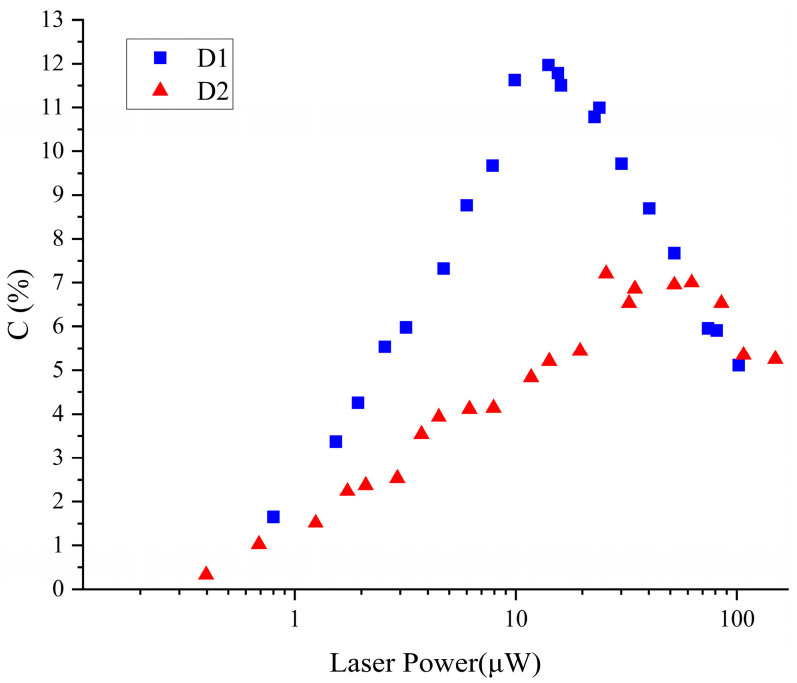
The relationship between CPT resonance contrast and the wavelength and laser power of a VCSEL.

**Figure 11 sensors-25-01229-f011:**
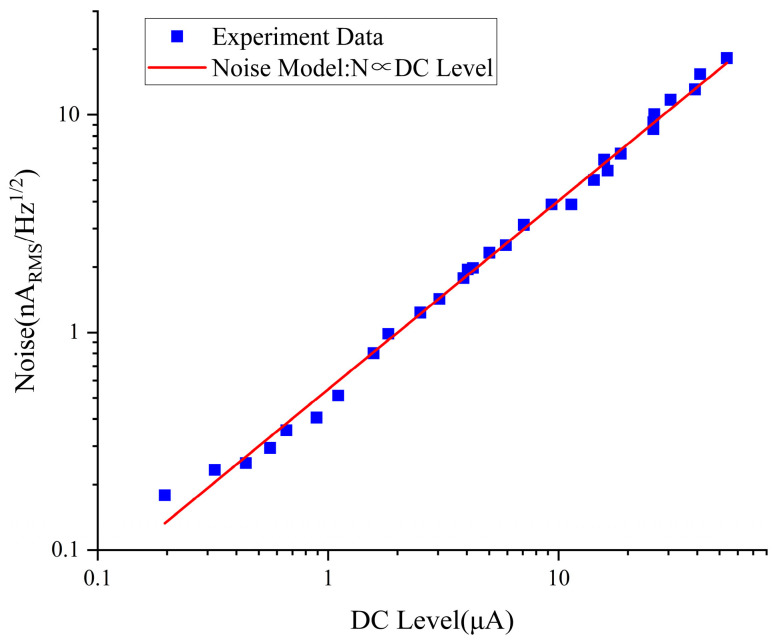
Laser noise power spectrum.

**Figure 12 sensors-25-01229-f012:**
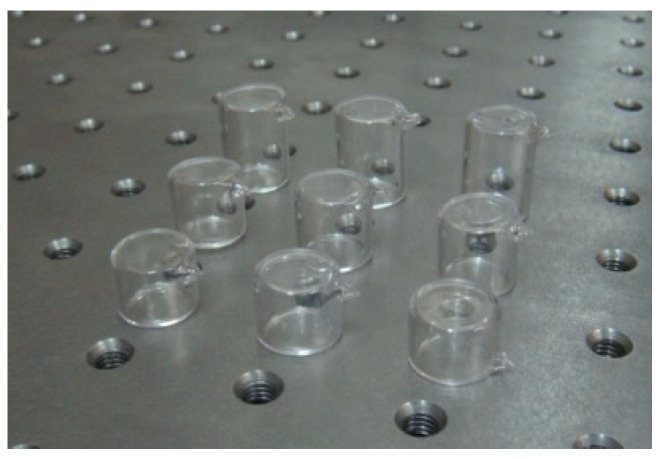
Different-sized cesium cells.

**Figure 13 sensors-25-01229-f013:**
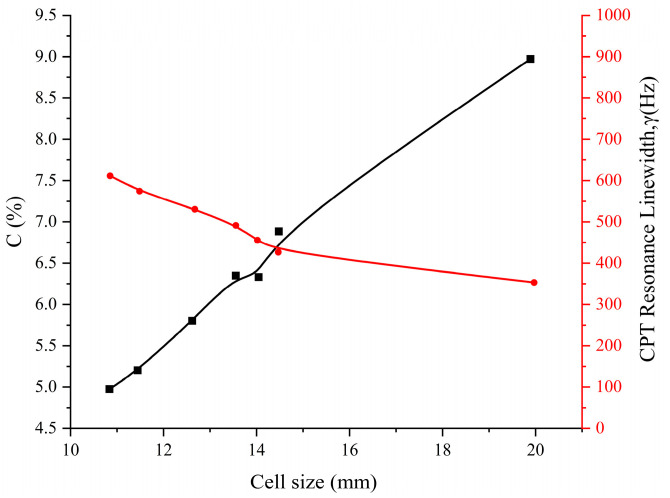
Relationship between different sizes of cesium cells and line width and contrast.

**Figure 14 sensors-25-01229-f014:**
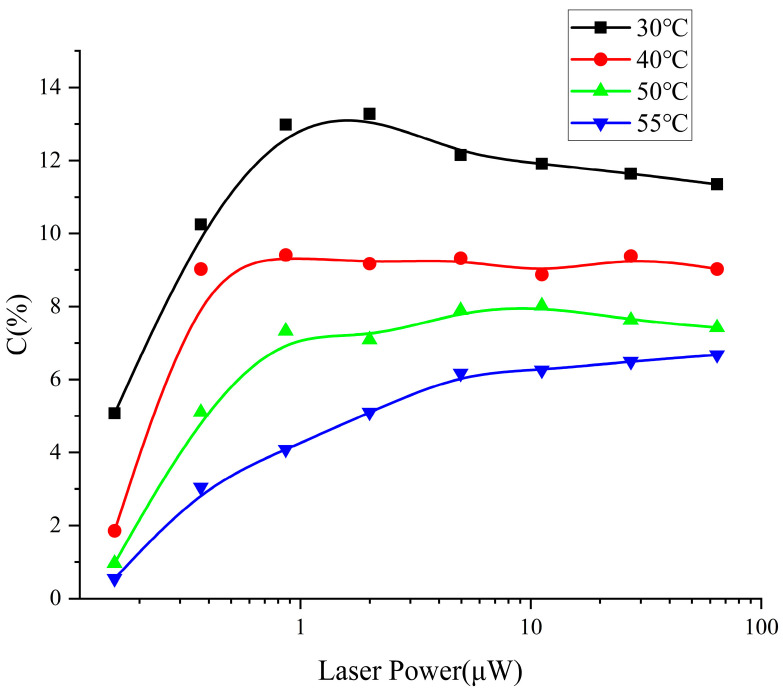
Contrast at different cesium cell temperatures and laser power.

**Figure 15 sensors-25-01229-f015:**
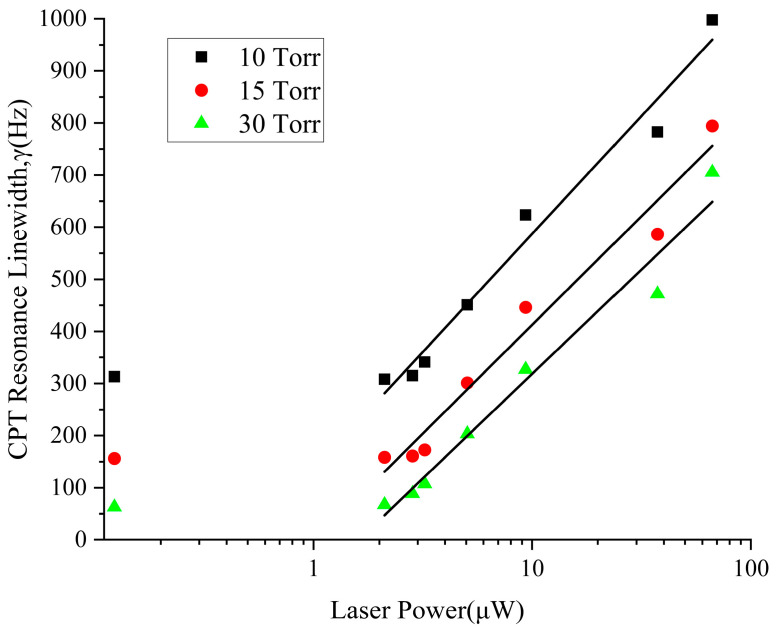
CPT Resonance signal linewidth and buffer gas pressure.

**Figure 16 sensors-25-01229-f016:**
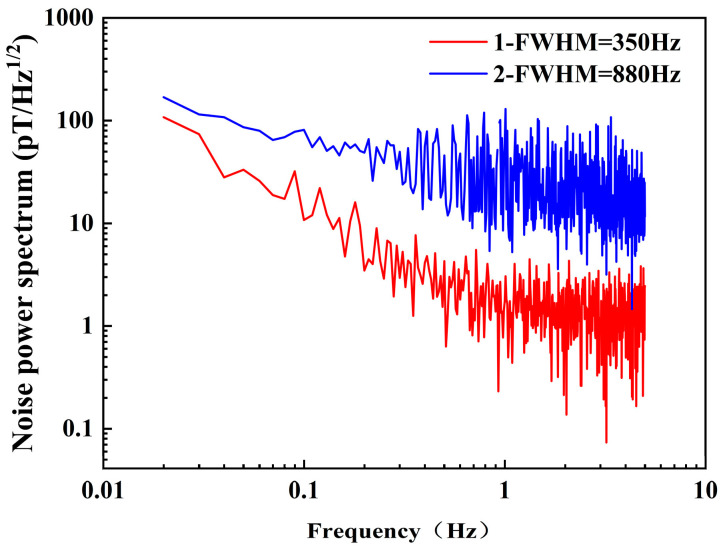
Improving the magnetic field measurement sensitivity.

**Table 1 sensors-25-01229-t001:** The pressure and temperature frequency shift coefficients for different buffer gasses.

Buffer Gasses	α (Hz/Torr)	δ (Hz/Torr) °C
N_2_	520 ± 13	0.6
Ne	392 ± 13	0.23
Ar	−51 ± 13	−0.3
CH	−500	−0.6

## Data Availability

The data presented in this study are available on request from the corresponding author.

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
