# Peer review of "Improving the Sensitivity of a Dark-Resonance Atomic Magnetometer"

_sensors, 2025, doi:10.3390/s25041229_

Round 1
Reviewer 1 Report
Comments and Suggestions for Authors
Dear Authors
I have a few questions:
1. Line 54. You presented a references [9,10,01]. Perhaps, instead of 01 should be 11 (?).
2. Line 79-85. "The Materials and Methods should be described with sufficient details to allow others to replicate and build on the published results. Please note that the publication of your manuscript implicates that you must make all materials, data, computer code, and protocols associated with the publication available to readers. Please disclose at the submission stage any restrictions on the availability of materials or information. New methods and protocols should be described in detail while well-established methods can be briefly described and appropriately cited."
Why did you insert the cited text into your paper?
3. Line 262-263. You wrote:"Both lasers have the 262 same 2mm beam diameter...".
Does it mean that you exposed only very small volume V of a cell (V=2mm x cell_length) in your experiments?
Did you investigate an influence of exposed volume on the sensitivity of your magnetometer?
4. Line 272. You mentioned the cell length, but did not mention the cell diameter. What was the diameter? How did you select the diameter value with respect to optimization of the sensitivity?
5. Line 349-350. "However, the noise will also increase accordingly, and the noise is not proportional to the temperature in a simple way."
What type of the noise did you mention here?
6. Line 401-403. "After optimizing the quantum system parameters, the magnetic field sensitivity of the coupled dark state magnetometer before and after the measurement was compared, as shown in Figure 16."
I do not understand the sentence. What measurement did you take into account here? How does the sensitivity depend on having the measurement done or not?
7. Did you compare your results (sensitivity) with the best sensitivity obtained by anothe reseachers for the same type of magnetometers? It would be very informative to present the comparison in you paper.
Best regards,
Reviewer
Reviewer 2 Report
Comments and Suggestions for Authors
The main contribution of the paper is the enhancement of the sensitivity of a dark state resonance atomic magnetometer, achieved through theoretical analysis and experimental setups that optimize parameters such as atomic cell size, buffer gas pressure, and operating temperature, resulting in a significant sensitivity improvement of approximately 20-fold. To further improve the study, the following suggestions can be proposed:
First, conducting a more extensive analysis of electronic noise sources, including the loop noise from various components, to identify specific factors that could be mitigated to enhance sensitivity.
Second, exploring alternative laser configurations or wavelengths beyond the D1 and D2 transitions to assess their potential impact on resonance contrast and linewidth.
Third, implementing a broader range of experimental conditions, such as varying environmental factors like temperature and pressure.
Fourth, incorporating advanced data analysis techniques, such as machine learning algorithms, to optimize the interpretation of experimental results and improve the reliability of sensitivity measurements.
Fifth, collaborating with other research groups to validate findings through independent experiments, which could strengthen the credibility of the results and facilitate the development of standardized protocols for future studies.
Reviewer 3 Report
Comments and Suggestions for Authors
The authors of this manuscript were extremely negligent in leaving template content in section 2.1, so I have given it a rejection. In addition, there are the following issues with this manuscript:
1. The meanings of all mathematical symbols need to be specified;
2. Reference 11 was mistakenly written as 01;
3. In references 9-11, no application of atomic magnetometer in UAV and UXO is found;
4. No clear conclusion is reached at the end of the introduction;
5. All sentences in the manuscript need further polishing.
6. In equation 3, V should be the interaction volume between the beam and the atoms, and S is the amplitude of which signal segment?
7. Please use specific function formulas to explain the difference between lines D1 and D2 in section 2.4.
8. Is the magnetic shielding device inside the oven in Figure 5?
9. What are the pressures of each component in the vapor cell in different experiments? What are the manufacturers and models of the equipment used for maintaining, stimulating, and detecting signals in the main testing environment? What is the data fitting algorithm?
10.Is the horizontal axis of Fig14 Laster? What does the vertical axis C % represent in Fig13 and Fig14?
11. The conclusion in Figure 15 needs to be supported by the profile formula and simulation.
12. This article improves sensitivity by manually adjusting experimental parameters, without proposing new methods or simulation evidence, and has almost no innovative points.
13. The unexploded bombs mentioned in the introduction were not involved in subsequent experiments. It is suggested to replace them with commonly used applications of magnetocardiography, electroencephalography, and geomagnetism, otherwise it may cause misunderstandings. All text in the figures should be in Times new roman type. Figure 12 should add the spatial scale. Why is nA used to represent laser power noise in Figure 11, and why is laser noise analyzed? as far as I know, the corresponding product manual already includes it.
Comments on the Quality of English LanguageThe language sentence structure is relatively simple, and the cause and effect are not coherent, requiring further polishing.
Round 2
Reviewer 2 Report
Comments and Suggestions for Authors
The authors considered my comments and suggestions. Good luck.
Reviewer 3 Report
Comments and Suggestions for Authors
1. The first question is still unresolved. What do E1, E2, ω 1, ω 2, ρ 1 and ρ2 mean in this article?, Why do some variables use italics inside the formula and regular fonts outside the formula? N2 requires subscripts, while γ ∝ P0.5 requires superscripts.
2. The sixth and ninth questions in the first review were not answered correctly, and the response to the twelfth question needs to be reflected in the original text. In Figure 15, it should be Torr, not TOrr. The ninth question emphasizes that the pressure of each component is not the total pressure.
3. The 'laster' in question 10 has been marked, why isn't the entire text changed to 'laser'?
4. If the oven is external, what are the magnetic shielding and coil support materials? Do they have good thermal conductivity?
5. The focus of the introduction is not on 'improving sensitivity', but rather on various applications.
6. The response to the seventh question further confirms that there is no noteworthy innovation in this article, as previous researchers have made a comprehensive theoretical/experimental comparison between the D1 and D2 lines.
Comments on the Quality of English LanguageLanguage needs further comprehensive polishing.
Round 3
Reviewer 3 Report
Comments and Suggestions for Authors
Apart from adding coordinate axis arrows in the figures and replacing them with clearer versions, there are no further questions.